# AN EXPERIMENT DESIGN PARADIGM USING JOINT FEATURE SELECTION AND TASK OPTIMIZATION

## ABSTRACT

This paper presents a subsampling-task paradigm for data-driven task-specific experiment design (ED) and a novel method in populationwide supervised feature selection (FS). Optimal ED, the choice of sampling points under constraints of limited acquisition-time, arises in a wide variety of scientific and engineering contexts. However the continuous optimization used in classical approaches depend on a-priori parameter choices and challenging non-convex optimization landscapes. This paper proposes to replace this strategy with a subsampling-task paradigm, analogous to populationwide supervised FS. In particular, we introduce JOFSTO, which performs JOint Feature Selection and Task Optimization. JOFSTO jointly optimizes two coupled networks: one for feature scoring, which provides the ED, the other for execution of a downstream task or process. Unlike most FS problems, e.g. selecting protein expressions for classification, ED problems typically select from highly correlated globally informative candidates rather than seeking a small number of highly informative features among many uninformative features. JOFSTO's construction efficiently identifies potentially correlated, but effective subsets and returns a trained task network. Changed: We demonstrate the approach using parameter estimation and mapping problems in clinically-relevant applications in quantitative MRI and in hyperspectral imaging. Results from simulations and empirical data show the subsampling-task paradigm strongly outperforms classical ED, and within our paradigm, JOFSTO outperforms state-of-the-art supervised FS techniques. JOFSTO extends immediately to wider image-based ED problems and other scenarios where the design must be specified globally across large numbers of acquisitions. Our code is available for reviewers Code (2022).

## 1 INTRODUCTION

Experiment design (ED) seeks an optimally informative sampling scheme within a budget of measurement time Antony (2003). The problem arises across a wide range of scientific disciplines and applications wherever mathematical models are fitted to resal-world noisy measurements to estimate quantities that cannot be measured directly e.g. agriculture Gupta et al. (2015), civil engineering Lye (2002), economics Jacquemet & L'Haridon (2019), microbiology Vanot & Sergent (2005). Classical approaches Frieden (2004); Montgomery (2001) optimize the design to minimize the uncertainty of parameter values of a prespecified model, often derived from the Fisher information matrix. For any non-linear model, these approaches require a-priori specification of model parameter values to optimize the design for, which leads to circularity, as parameter values are by definition unknown at application stage. Moreover the optimization itself is usually cumbersome over a high-dimensional and highly non-convex space.

For example, quantitative imaging techniques estimate and map model parameters pixel by pixel from multi-channel images. Multiple acquisition parameters often control the contrast in each channel. The ED challenge is to identify the combination of acquisition parameters that best inform the estimation of the model parameters, which vary substantially over the image. The popular MRI brain-imaging technique NODDI exemplifies the challenges: five acquisition parameters can vary for each of around 100 channels, thus the ED optimization is 500 dimensional. The standard acquisition protocol was designed by optimizing the Fisher-matrix for one specific combination of parameter values, although the aim of the technique is to highlight contrast in those parameters over the extent of the brain - the acquisition protocol is therefore by definition suboptimal.

This paper considers a new paradigm for ED, which we cast as a populationwide feature selection (FS) problem, instead of one of continuous optimization. The paradigm requires training data densely sampled over the space of possible measurements from a representative population of test cases. It identifies i) a subset of locations in the measurement space that allows high performance of a downstream task, and ii) a trained network to support this task. The problem differs from most FS problems addressed by recent approaches Lee et al. (2022); Wojtas & Chen (2020) which use e.g. in protein-coding genes or noisy two-moons data, which typically aim to 'identify a small, highly discriminative subset' Kuncheva et al. (2020). Typically in ED, each measurement individually offers similar amounts of information overall to support task performance, i.e. ED measurements are all informative, but informs different aspects of the task; ED seeks a combination that covers all important aspects. Accordingly, again unlike most FS problems, measurements are highly correlated (see figure 3).

Thus we propose JOFSTO: an approach for joint FS and task optimization, applied to task-driven ED. JOFSTO's novel architecture simultaneously trains two neural networks end-to-end; the first scores and ranks the features by relevance to optimize subsampling and the second uses a subset of measurements to perform a prespecified task, such as estimating ED model parameters. JOFSTO's subsampling coupled with downstream-supported task training outputs both an optimized ED and a network trained for optimal task performance given that design. A simple scoring mechanism enables JOFSTO to identify effective subsets of potentially highly correlated features efficiently, and joint training during subsampling maintains high task performance. Furthermore we put JOFSTO's novel approach to scoring and feature subsampling in a recursive feature elimination (RFE) framework, to reduce the full set of samples to a small subsample stepwise, which improves the optimization and aids convergence to strong solutions.

We demonstrate the benefits of JOFSTO within applications in quantitative (MR and hyperspectral) imaging, where the standard aim is to fit a model to multichannel measurements in each image voxel to obtain informative parameters that provide e.g. biological information. In MRI, acquisition time is limited by factors of cost and the ability of subjects to remain motionless in the noisy and claustrophobic environment of the scanner, thus ED is crucial to support the most accurate image-driven diagnosis, prognosis, or treatment choices. In hyperspectral imaging, recovering high-quality information from the few wavelengths chosen by ED, increases acquisition speed, avoids misalignment, reduces storage requirements, and speeds up clinical adoption. Experiments, using both simulations and real-world data, show that JOFSTO outperforms classical ED and produces state-of-the-art performance in a subsampling-reconstruction MRI challenge, and oxygen saturation estimation. Moreover, within the subsampling paradigm, JOFSTO outperforms state-of-the-art FS algorithms Wojtas & Chen (2020); Lee et al. (2022) on four datasets/tasks.

## 2 RELATED WORK

**Experiment Design (ED)** The design of an experiment is a set $A = \{\mathbf{a}^1, ..., \mathbf{a}^C\}$ where each $\mathbf{a}^i \in A$ is a combination of acquisition-parameter settings, i.e. choices of independent variables under the control of the experimenter and $C$ is the number of measurements acquired. Each data acquisition under ED $A$ provides a set of measurements $\mathbf{x} = (x^1, ..., x^C)$, corresponding to the elements of $A$; $\mathbf{x}$ is a *sample* under design $A$. ED optimization seeks the $A$ that maximally supports a task, such as estimating parameters $\boldsymbol{\theta}$ of model $f(\mathbf{x}; \boldsymbol{\theta})$, that relates measurements, $\mathbf{x}$, of a system to underlying properties of interest encapsulated in $\boldsymbol{\theta}$. Classical ED approaches typically aim to minimize the expected variance of model parameters e.g. encoded in the Fisher information matrix Pukelsheim (2006), using Bayesian techniques Chaloner & Verdinelli (1995); Kaddour et al. (2020) or indirectly via an operating characteristic curve Montgomery (2001). However, for non-linear models, parameter uncertainty depends on parameter value, so the design-optimization requires pre-specification of parameter values of interest, leading to circularity. Moreover, the optimization often becomes cumbersome particularly as model complexity increases and $C$ becomes large Alexander (2008). Our subsampling-task paradigm for ED avoids this circularity and replaces the challenging continuous optimization problem with neural network training. Recent work e.g. Pizzolato et al. (2020); Blumberg et al. (2022); Waterhouse & Stoyanov (2022) raise the possibility of a subsampling approach to ED, but consider it only in specific scenarios, whilst Grussu et al. (2021) does not densely-sample the measurement and spatial domains, instead arguing that this leads to a model-free ED. Here we uniquely couple the subsampling with specific downstream tasks to optimize jointly the

execution of a parameter mapping or other downstream task with identification of the design that optimally supports it.

**Supervised Feature Selection (FS)** FS is the well-studied problem of identifying an informative subset of features Guyon et al. (2004); Liu & Motoda (1998), either at the instance level e.g. identifying different salient parts of different images; or at the population/global level selecting across all the instances. Populationwide supervised FS is of primary relevance here, since each combination of acquisition parameters, i.e. elements of $A$, is global to many samples (image voxels). State-of-the-art approaches Wojtas & Chen (2020); Lee et al. (2022) outperform classical approaches e.g. (original) RFE Guyon et al. (2002), BAHSIC Song et al. (2007; 2012), mRMR Peng et al. (2005), CCM Chen et al. (2017), RF Breiman (2001), DFS Li et al. (2016), LASSO Tibshirani (1996), L-Score He et al. (2005); and recently, DL-based Abid et al. (2019); Yamada et al. (2020); Lindenbaum et al. (2021).

**Strategies for Supervised FS** There are two strategies for populationwide supervised FS, which aim to 'identify a small, highly discriminative subset' Kuncheva et al. (2020). The first strategy e.g. Lee et al. (2022) first exploits elements of task-performance, self-supervision, additional unlabeled data, and correlated feature subsets, to score all features, then perform FS for chosen subset size $C$. In a second step, a network is trained on feature subset for the task. However, this step does not exploit information from all of the features – assumed to be unnecessary for e.g. classification on DNA/RNA/proteins, or even counterproductive when a large number of nuisance features masks the structured features e.g. noisy two-moons dataset. This differs from our ED paradigm, where such a discriminative subset may not exist, as data are globally informative and e.g. has large global correlation (figure 3). A second strategy combines training for FS and task optimization, and returns an optimal feature subset of pre-given size $C$ and a trained task network. For example, Wojtas & Chen (2020) has unconnected dual feature scoring and task-prediction networks and a stochastic local search. However in the ED paradigm, this unnecessarily searches for the discriminative subset, which involves the NP hard problem of assessing input-target functional dependence Weston et al. (2003) and performing multiple task evaluations on different feature combinations. This may result in overfitting and long training times. Here we follow the second strategy with jointly optimized and connected dual networks, but avoid searching for the highly-discriminative subset.

## 3 METHODS

Figure 1 outlines JOFSTO's computational graph with the example tasks we demonstrate later. This section gives a conceptual overview of the procedure. Section A, in particular Algorithm 2, defines the details of the implementation. The $\cdot$ operation is element-wise and follows broadcasting rules.

**Overview** JOFSTO presents a novel approach to supervised FS, by performing concurrent feature scoring, feature subsampling, and task-prediction. In an outer loop, JOFSTO considers decreasing feature set sizes, instead of e.g. subsampling all required features at once, improving the optimization procedure. It also transfers information from larger feature sets to smaller ones. For a fixed step in the outer loop, JOFSTO trains a novel dual-network in an inner loop, where the networks score the features and performing the task. As training is performed alongside a novel approach in progressive feature elimination, this maintains high predictive performance during subsampling.

**Problem Specification** Our subsampling ED paradigm requires an input data set $\bar{X} = \{\bar{\mathbf{x}}_1, ..., \bar{\mathbf{x}}_n\} \in \mathbb{R}^{n \times \bar{C}}$, which consists of $n$ samples and $\bar{C}$ features acquired under super-design $\bar{A} = \{\bar{\mathbf{a}}^1, ..., \bar{\mathbf{a}}^{\bar{C}}\}$ that densely-samples the measurement/feature space. We also require a target data set $Y = \{\mathbf{y}_1, ..., \mathbf{y}_n\} \in \mathbb{R}^{n \times M}$, which provides gold-standard output of the task driving the ED; each $\mathbf{y}_i$ is a vector of $M$ targets (e.g. MRI model parameters) for the corresponding sample $\bar{\mathbf{x}}_i$. JOFSTO aims to subsample $C < \bar{C}$ acquisition-parameter combinations, corresponding to $C$ features, to obtain an economical design $A = \{\mathbf{a}^1, ..., \mathbf{a}^C\} \subset \bar{A}$, which corresponds to acquiring $X = \{\mathbf{x}_1, ..., \mathbf{x}_n\} \in \mathbb{R}^{n \times C}$ where the same $C$ elements of each $\bar{\mathbf{x}}_i$ are in $\mathbf{x}_i$.

**Outer Loop** JOFSTO considers multiple candidates for $C$ across steps $t = 1, ..., T$ and gradually samples from the full data set of $\bar{C}$ features to a minimum candidate of $C_T$, i.e. the user selects hyperparameters $C_t$, $t = 1, ..., T$, $\bar{C} = C_1 > C_2... > C_T$. At each step $t$, JOFSTO aims to construct a mask to subsample the features $m_t \in \{0, 1\}^{\bar{C}}, ||m_t|| = C_t$ so $X = m_t \cdot \bar{X} + (1 - m_t) \cdot \bar{X}^{fill}$ where $\bar{X}^{fill} \in \mathbb{R}^{\bar{C}}$ is user-chosen and, a score for the features $\bar{s}_t \in \mathbb{R}^{\bar{C}}_+$, which is also used to

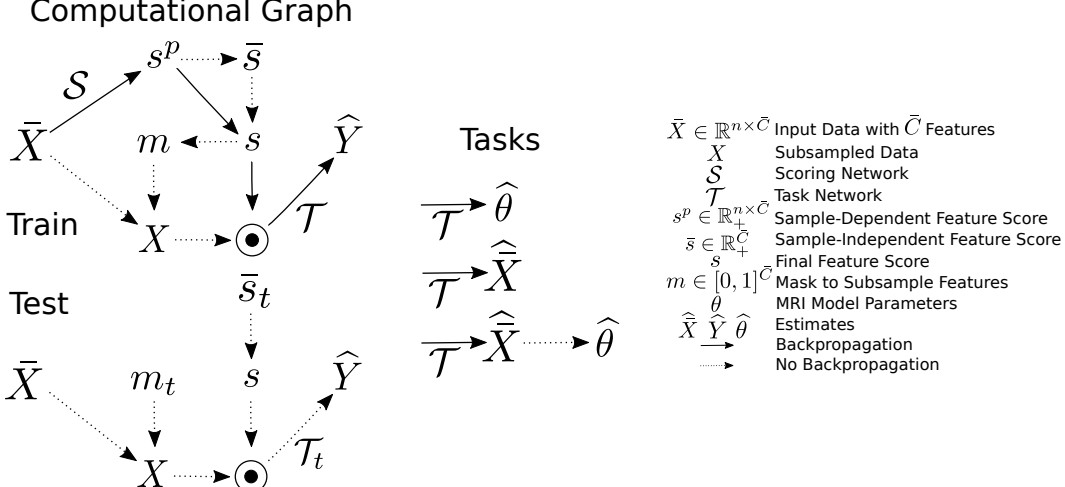

Figure 1: JOFSTO trains two neural networks $\mathcal{S}, \mathcal{T}$ end-to-end, where gradients are passed through the black lines. The score $s$ for each measurement is a combination of learnt score $s^p$, and an averaged score $\bar{s}$ that is not directly learnt. During training in step $t$, JOFSTO progressively sets $s$ to $\bar{s}$ to construct $\bar{s}_t$, and progressively constructs the mask $m_t$ to subsample the data, $\bar{X}$, based on the lowest scores. We consider three types of tasks with distinct target output $\widehat{Y}$: mapping each sample to a set of model parameters in sections 4.1.1,4.2.2; reconstructing the full data set in sections 4.1.24.2.1; estimating model parameters from the reconstructed full data set in section 4.1.3.

normalize the features, and a trained task network $\mathcal{T}_t$, to minimize $||Y - \mathcal{T}_t(X \cdot \bar{s}_t)||$. As the trained network $\mathcal{T}_t$ is used to construct $\mathcal{T}_{t+1}$, information is passed from larger feature sets to smaller ones. JOFSTO gradually reduces the feature set sizes across $t = 1, ..., T$ (instead of e.g. subsampling all the features at once), which also improve the optimization procedure. JOFTO's novel approach to scoring and feature subsampling is a novel method in the RFE framework Guyon et al. (2002), itself in the backward elimination wrapper framework Kohavi & John (1997).

**Computational Graph and Network** JOFSTO has two neural networks: a Scoring Network $\mathcal{S}$ to score the features and a Task Network $\mathcal{T}$ to perform the task, and they are trained consecutively and end-to-end, so feature scoring is concurrent to task prediction. First JOFSTO learns a data-dependent score $s^p = \sigma(\mathcal{S}(\bar{X}))$ which in practise is computed across batches, where $\sigma : \mathbb{R} \to [0, \infty)$ is an activation function to ensure positive scores. The final feature score $s$ is a linear combination of the learnt score $s^p$, and a sample-independent score $\bar{s}$ which is an average of $s^p$ across the entire data and a candidate for $\bar{s}_t$. This is expressed as $s = \alpha^s \cdot s^p + (1 - \alpha^s) \cdot \bar{s}$, where $\alpha^s \in [0, 1]$ is set by the user. JOFSTO uses a mask $m \in [0, 1]^{\bar{C}}$ to subsample the measurements with $X = m \cdot \bar{X} + (1 - m) \cdot \bar{X}^{fill}$ where $\bar{X}^{fill} \in \mathbb{R}^{\bar{C}}$ is user-chosen (e.g. zeros or data median) and fills the locations of un-subsampled measurements. The Task Network $\mathcal{T}$ takes the weighted, subsampled measurements and estimates the target $\widehat{Y} = \mathcal{T}(s \cdot X)$, where multiplication here allows gradients to flow end-to-end. A supervised loss is calculated $L(\widehat{Y}, Y)$, for user-chosen $L$, allowing computation of gradients of $\mathcal{S}, \mathcal{T}$.

**Progressive Network Modification** JOFSTO's construction of $m_t, \bar{s}_t$ is motivated by previous work Karras et al. (2018); Blumberg et al. (2019). These approaches progressively modified neural network topology across training, from learning a simpler task to a more complicated task.

**Inner Loop Training** In step $t$, JOFTO performs standard deep learning training in the inner loop. As DL training is on batches, but we want to learn and then train on a sample-independent score, we first learn a sample-dependent score $s^p \in \mathbb{R}_+^{n \times \bar{C}}$, which is progressively set to its average $\bar{s}$ across the batches. This is done by by progressively setting $\alpha^s$ from 1 to 0 which sets $s = s^p$ to $s = \bar{s}$. Then, $\bar{s} \in \mathbb{R}_+^{\bar{C}}$ ranks the $\bar{C}$ features and the $C_{t-1} - C_t$ lowest-scored features are chosen to be removed. $m$ is progressively modified to have $C_{t-1}$ ones and $\bar{C} - C_{t-1}$ zeros, to having $C_t$ ones and $\bar{C} - C_t$ zeros during training, so the dual networks adapt to subsampling and maintain high task performance with

Table 1: Mean-Squared-Error $\times 10^2$ between estimated model parameters and ground truth model parameters used to simulate data, comparing JOFSTO with classical Fisher Matrix ED from the literature.

| | | VERDICT | | NODDI |
|---|---|---|---|---|
| Classic ED | 15.0 | $C = 20$ | 8 | $C = 99$ |
| JOFSTO | **2.04** | $C = 20, \bar{C} = 220$ | **4.51** | $C = 99, \bar{C} = 3612$ |

Table 2: Mean-Squared-Error $\times 10^2$, between estimated model parameters and ground truth model parameters used to simulate data, comparing JOFSTO against approaches in supervised feature selection, selecting $C$ from $\bar{C}$ measurements

| | VERDICT $\bar{C} = 220$ | | | | NODDI $\bar{C} = 3612$ | | | |
|---|---|---|---|---|---|---|---|---|
| $C =$ | 110 | 55 | 28 | 14 | 1806 | 903 | 452 | 226 |
| Random FS + DL | 1.54 | 2.24 | 3.25 | 6.10 | 2.99 | 3.34 | 3.88 | 4.31 |
| Lee et al. (2022) | 1.06 | 1.28 | 1.89 | 4.58 | 2.95 | 3.39 | 3.73 | 4.39 |
| Wojtas & Chen (2020) | 2.22 | 2.14 | 3.09 | 4.05 | 4.21 | 4.61 | 4.96 | 5.14 |
| JOFSTO | **1.03** | **1.18** | **1.80** | **2.64** | **2.59** | **2.92** | **3.33** | **3.85** |

continual training. Further details are in algorithm 2. At the completion of step $t$, we take $m_t, \bar{s}_t, \mathcal{T}_t$, as $m, \bar{s}, \mathcal{T}$ i.e. figure 1-Test.

## 4 EXPERIMENTS AND RESULTS

This section presents the experiments and results in the example applications of qMRI in section 4.1 and hyperspectral imaging in section 4.2. Table F.3 analyzes the impact of removing JOFSTO's components on performance and examines how the random seed impacts performance.

**Data** Data contains densely-sampled measurements in the acquisition parameter space, produced either by scanning in-vivo human brains, via simulations using qMRI models, or from real and simulated hyperspectral images, and are visualized in section F. Here, each element $\bar{\mathbf{a}}$ of the super-design ED $\bar{A}$ produces a 2D/3D grayscale image, so $\bar{A}$ correspond to 2D/3D images with $\bar{C}$ channels. The econonmical design $A$ then corresponds to a subset of the image channels. Following standard practice in MR parameter estimation and mapping Cercignani et al. (2018); Alexander et al. (2019), and hyperspectral image filter design Waterhouse & Stoyanov (2022), the data samples $\mathbf{x}_i \in \mathbb{R}^{\bar{C}}$ are individual voxels, thus the $\bar{C}$ features are the measurements/channels. As the statistical distribution of voxels in each measurement/channel is different, we normalize the data channel-wise at the input and output of all neural networks for more efficient training.

**Baselines** Experiments evaluate against two levels of baseline. First, to evaluate the subsampling ED paradigm in general, we compare JOFSTO to classical qMRI ED techniques using the Fisher information matrix Alexander (2008). Then, within the subsampling paradigm, we consider three baseline supervised FS methods. Firstly, random FS then deep learning (DL) training – an off-the-shelf, simple approach; and two baselines described in section 2: Lee et al. (2022) – the state-of-the-art approach in supervised FS, and Wojtas & Chen (2020) which, like JOFSTO has dual feature scoring and task prediction networks. Each approach conducts a brief hyperparameter search, and for fairness, the same number of evaluations are used for each feature subset. We used the official code repository for the baselines when required. Further details are in supplementary section B.

### 4.1 APPLICATION TO QUANTITATIVE MAGNETIC RESONANCE IMAGING (qMRI)

This section presents results on four experiments, to show that JOFSTO has enhanced capacity, compared to alternative approaches, for the ED problem of designing an efficient MRI acquisition scheme $A$ underpinning clinically useful tasks. Section 4.1.1 examines learning ground truth model parameters from simulations of two widely used qMRI models, that map histologic features of tumors and complexity of neural dendrites and axons. Section 4.1.2 explores a MRI subsampling-

Table 3: Mean-squared-error between $\bar{C} = 1344$ reconstructed measurements and $\bar{C}$ ground-truth measurements, on same experimental settings as Blumberg et al. (2022).

| | $C = 500$ | 250 | 100 | 50 | 40 | 30 | 20 | 10 |
|---|---|---|---|---|---|---|---|---|
| Blumberg et al. (2022) Table 1 State-of-Art | 0.49 | 0.61 | 0.89 | 1.35 | 1.53 | 1.87 | 2.50 | 3.48 |
| JOFSTO | **0.22** | **0.43** | **0.88** | **1.34** | **1.52** | **1.76** | **2.12** | **2.88** |

Table 4: Mean-squared-error between $\bar{C} = 1344$ reconstructed measurements and $\bar{C}$ ground-truth measurements, on MUlti-DIffusion (MUDI) challenge subjects.

| | $C = 500$ | 250 | 100 | 50 |
|---|---|---|---|---|
| Random FS + DL | 0.93 | 1.41 | 2.12 | 5.23 |
| Lee et al. (2022) | 0.63 | 0.86 | 1.24 | 1.61 |
| Wojtas & Chen (2020) | 1.67 | 1.72 | 2.17 | 2.34 |
| JOFSTO | **0.21** | **0.44** | **0.94** | **1.34** |

reconstruction challenge. Section 4.1.3 focusses on estimating multiple clinically relevant downstream metrics, computed by fitting qMRI models on reconstructed data.

### 4.1.1 JOFSTO PRODUCES BETTER PARAMETER ESTIMATES THAN BASELINES ON SIMULATIONS

This section shows JOFSTO outperforms classical approaches in ED and in supervised FS, in a setting where only simulated data is accessible, to best estimate tissue-related model parameters. This demonstrates that JOFSTO can learn time-efficient qMRI protocols without any MRI scans

Recent trends in qMRI Palombo et al. (2020); Gyori et al. (2022) use DL to estimate ground truth (biophysical) model parameters. We adopt that idea with two qMRI models: VERDICT which aids early detection and classification of prostate cancer Panagiotaki et al. (2015a), and NODDI, which has been applied in Alzheimer's disease Kamiya et al. (2020) and multiple sclerosis Grussu et al. (2017). Both models are described in section C. We use simulations with known ground truth parameters to evaluate the different ED approaches via accuracy of model-parameter estimation. For the classical ED approach, we use two published Fisher matrix-optimized acquisition schemes for VERDICT from Panagiotaki et al. (2015b) where $C = 20$; and NODDI from Zhang et al. (2012) where $C = 99$. For the supervised FS approaches, we select features from model-specific densely-sampled acquisition schemes for VERDICT ($\bar{C} = 220$) Panagiotaki et al. (2015a); and NODDI ($\bar{C} = 3612$) Ferizi et al. (2017). Simulated input qMRI data is created with $\mathbf{x}_i = f_{\boldsymbol{\theta}_i}(\mathbf{a}) + noise$, for MRI model $f$, with parameters $\boldsymbol{\theta}_i$ at sample/voxel $i = 1, ..., n$, generated from a biologically plausible range and are fixed across different experiments, described in section F, and Rician noise Gudbjartsson & Patz (1995) representative of clinical qMRI. Target data are the ground truth model parameters $Y = \{\boldsymbol{\theta}_1, ..., \boldsymbol{\theta}_n\}$. Performance is the MSE between ground truth and learnt parameters across the voxels.

Table 1 compares the classical Fisher information matrix ED Alexander (2008) against the JOFSTO, where JOFSTO FS size $C$ is set to the number of measurements produced by the classical ED acquisition scheme. Table 2 shows JOFSTO against alternative approaches in supervised FS, where $C = \frac{\bar{C}}{2}, \frac{\bar{C}}{4}, \frac{\bar{C}}{8}, \frac{\bar{C}}{16}$. Further details regarding the models, experimental settings, and data are in sections F. In both cases, JOFSTO outperforms the baselines.

### 4.1.2 JOFSTO PRODUCES STATE-OF-THE-ART PERFORMANCE IN THE MUDI RECONSTRUCTION CHALLENGE

This section shows that JOFSTO obtains state-of-the-art performance, and outperforms approaches in supervised FS, in a subsampling-reconstruction MRI challenge.

The MICCAI MUlti-DIffusion (MUDI) challenge Pizzolato et al. (2020) aimed to identify an informative subset of $C = \{500, 250, 100, 50\}$ measurements from a densely sampled qMRI data set with $\bar{C} = 1344$ measurements, then use the subset to reconstruct the full data set i.e. $Y := \bar{X}$. This reconstruction task first provides a generic challenge that tests the ability of an ED or FS algorithm to identify a subset with maximal information content. The MUDI data detailed in section F, comes from a state-of-the art qMRI technique that acquires multiple MRI modalities simultaneously in a

Table 5:  Mean-squared-error between parametric maps (downstream metrics) estimated from $\bar{C} = 288$ ground truth measurements and $\tilde{C}$ from $C$ reconstructed measurements.

| $C = 18$ | DTI | | | | DKI | | | MSDKI | |
|---|---|---|---|---|---|---|---|---|---|
| | FA | MD | AD | RD | MK | AK | RK | MSD | MSK |
| Random FS + DL | 2.23 | 6.09 | 22.7 | 6.97 | 9.03 | 7.83 | 15.3 | 6.82 | 7.59 |
| Lee et al. (2022) | 2.86 | 12.9 | 31.2 | 14.9 | 12.0 | 9.26 | 20.3 | 8.96 | 8.17 |
| Wojtas & Chen (2020) | 9.83 | 23.2 | 77.7 | 26.8 | 11.9 | 10.9 | 21.3 | 10.8 | 6.03 |
| JOFSTO | **1.29** | **2.55** | **13.4** | **2.60** | **7.67** | **6.73** | **13.9** | **6.37** | **4.94** |

Figure 2:  Qualitative comparison of 2D slices from 3D parametric maps (downstream metrics), where gold standard is estimated from $\bar{C} = 288$ ground truth measurements, and baseline/ours is from $C = 18$ reconstructed measurements. JOFSTO's parametric maps are visually closer to the gold standard than those from the best performing baseline.

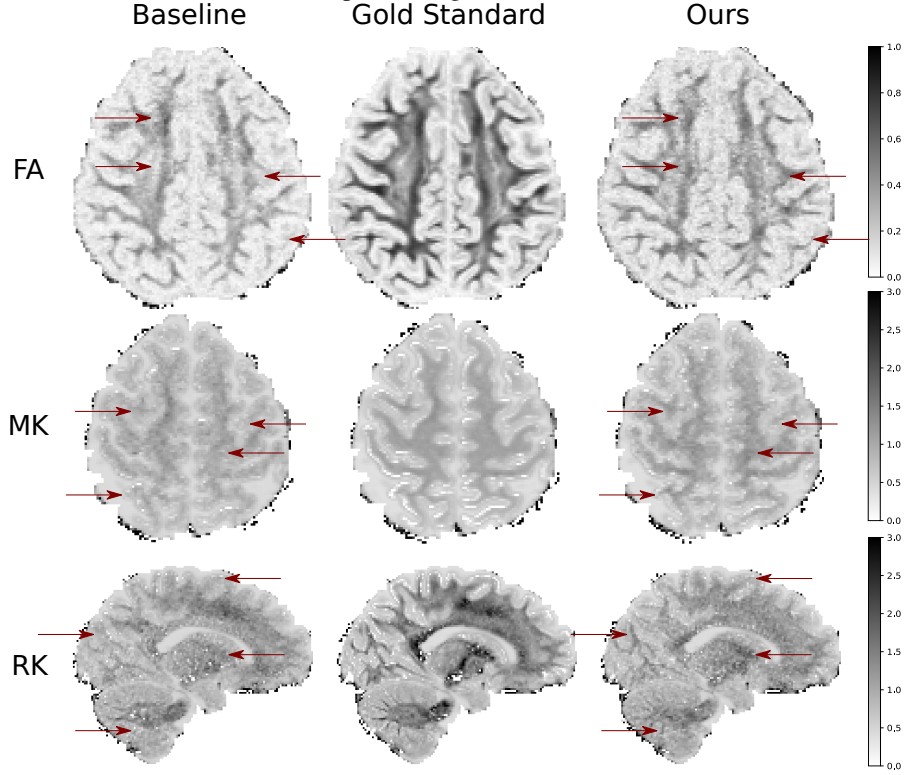

high-dimensional acquisition parameter space where $A \subset \mathbb{R}^6$, underscoring the importance of ED, as adequately sampling this high-dimensional measurement space in a time budget that is realistic in clinical settings is difficult Slator et al. (2021). We consider two experiments that highlight the benefits of JOFSTO in this paradigm.

The first experiment follows the setting in Blumberg et al. (2022), which has state-of-the-art performance on the data, table 3 shows results, confirming that JOFSTO improves over the state-of-the-art. The second experiment compares JOFSTO with the state-of-the-art techniques in supervised FS in the settings of the original challenge, and table 4 shows that JOFSTO outperforms these techniques.

### 4.1.3    JOFSTO IMPROVES ESTIMATION OF TISSUE MICROSTRUCTURE FROM qMRI SCANS

This section shows JOFSTO outperforms approaches in supervised FS for the ED task of designing an efficient acquisition scheme to support reconstruction of a densely-sampled dataset and downstream in-vivo quantification of human tissue microstructure.

Table 6: Mean-squared-error between $\bar{C} = 220$ reconstructed measurements and $\bar{C}$ ground-truth measurements, on Indian Pine AVIRIS hyperspectral data.

| | $C = 110$ | 55 | 28 | 14 |
|---|---|---|---|---|
| Random FS + DL | 1.22 | 1.76 | 2.91 | 5.61 |
| Lee et al. (2022) | 1.36 | 3.11 | 3.77 | 7.61 |
| Wojtas & Chen (2020) | 6.34 | 6.68 | 7.16 | 7.78 |
| JOFSTO | **0.60** | **1.42** | **2.33** | **4.49** |

Table 7: Root-Mean-Squared-Error $\times 10^2$ between the predicted and ground truth oxygenation (abundance maps), same experimental settings as Waterhouse & Stoyanov (2022).

| | $C = 6$ | 5 | 4 | 3 | 2 |
|---|---|---|---|---|---|
| Waterhouse & Stoyanov (2022) Fig. 5a Best | 4.54 | 4.91 | 5.33 | 6.17 | 7.55 |
| JOFSTO | **3.22** | **3.61** | **4.64** | **4.94** | **5.18** |

DTI Basser et al. (1994), DKI Jensen & Helpern (2010), and MSDKI are widely-used qMRI methods, described in detail in section D, that can quantify tissue microstructure, and show promise for extracting imaging biomarkers for many medical applications, such mild brain trauma, epilepsy, stroke, and Alzheimer's disease Ranzenberger & Snyder (2022); Tae et al. (2018); Jensen & Helpern (2010). This experiment uses rich, high-resolution HCP data detailed in section F with $\bar{C} = 288$. The task is to subsample feature set sizes $C = \frac{\bar{C}}{8}, \frac{\bar{C}}{16}$ then reconstruct the data, where the models are then fitted using standard techniques detailed in section D. Therefore the aim is to reduce the acquisition requirements for these imaging techniques, thereby enabling their usage in a wider range of clinical application areas. Quantitative and qualitative results are in table 5,supplementaries-table 8,figure 2, where the JOFSTO outperforms the baselines quantitatively in 17/18 comparisons on clinically useful downstream metrics (parameter maps) and JOFSTO's parametric maps are visually closer to the gold standard than those from the best baseline.

## 4.2 APPLICATION TO HYPERSPECTRAL IMAGING

Hyperspectral imaging (a.k.a. imaging spectroscopy) obtains pixel-wise information across multiple wavelengths of the electromagnetic spectrum from specialized hardware, recovering detailed information of the examined object-of-interest Manolakis et al. (2016). For example, the JPL's Airborne Visible / Infrared Imaging Spectrometer (AVIRIS) Thompson et al. (2017) remotely senses elements of the Earth's atmosphere and surface from airplanes, and has been used to examine the effect and rehabilitation of forests affected by large wildfires. This produces an 'image cube' - a 2D spatial image with multiple channels, where, similar to qMRI, the image channels correspond to the measurements. In this paradigm, ED consists of choosing the wavelengths and/or filters Arad & Ben-Shahar (2017); Waterhouse & Stoyanov (2022); Wu et al. (2019), which controls the number of channels. The aim is to obtain high-quality information from only a few measurements (corresponding to image channels), which can reduce the cost of current acquisitions or spur the development of more economical hardware Stuart et al. (2020). We conduct two experiments with hyperspectral data, subsampling then reconstructing AVIRIS images, and estimating oxygen saturation (a model parameter-estimation task).

### 4.2.1 JOFSTO IMPROVES ON SUBSAMPLING-RECONSTRUCTING HYPERSPECTRAL IMAGES

This experiment follows the approach in section 4.1.2 and examines a sampling-reconstruction task on publicly-available, high-quality AVIRIS ground images. To support soils research, Purdue University Agronomy obtained AVIRIS data from a north-to-south flight, dubbed the 'Indian Pine' dataset Baumgardner et al. (2015), where $\overline{C} = 220$ corresponds to the number of wavelengths. Results in table 6 show that JOFSTO improves over state-of-the-art approaches in supervised FS, for the subsample sizes $C = \frac{\bar{C}}{2}, \frac{\bar{C}}{4}, \frac{\bar{C}}{8}, \frac{\bar{C}}{16}$.

### 4.2.2 JOFSTO Improves the Estimation of Oxygen Saturation

This experiment is in the setting of Waterhouse & Stoyanov (2022). This examines how to design spectral filter sets to allow the estimation of oxygen saturation via obtaining non-invasive and non-destructive hyperspectral images, via the model in Can & Ülgen (2019). In this setting, the measurements are obtained from a filter applied to a wavelength (center), producing $\tilde{C} = 4 \cdot 87$ measurements, and the target is 2-dimensional i.e. $M = 2$. Results in table 7 show that JOFSTO outperforms the six approaches in Waterhouse & Stoyanov (2022) for feature sizes $C = 6, 5, 4, 3, 2$.

## 5 Discussion

This paper proposes to replace classic approaches in ED with a subsampling-task paradigm analogous to populationwide supervised FS, thereby avoiding complications in continuous optimization and dependence of the design on a-priori model parameter choices. We introduce JOFSTO, which performs JOint Feature Selection and Task Optimization, that takes advantage of the structure of ED data. Experiments address the clinically relevant problem of obtaining high-quality microstructural information in the limited time available in MRI scans, and hyperspectral imaging. Results show JOFSTO outperforms classical ED and state-of-the-art approaches in populationwide supervised FS. JOFSTO also obtains state-of-the-art performance in a subsampling-reconstruction MRI challenge and in oxygen saturation. The code is available to reviewers Code (2022).

Our paradigm assumes we can obtain densely-sampled data in the measurement space. This is possible in MRI, as only a few subjects are required to provide densely-sampled training data. They may be e.g. sedated, and remain motionless for the long acquisition times required. Whilst such data is acquirable in $\approx 1$ hour in research studies, in extreme cases motivated researchers endured longer, e.g. 8 hours in Ferizi et al. (2017). Similarly, in hyperspectral imaging, expensive devices can acquire large numbers of images with different spectral sensitivity simultaneously to provide training data for the design of much cheaper deployable devices with a small but informative set of sensitivities for particular applications. The same paradigm is applicable in other imaging problems and beyond, e.g. studies of cell populations Sinkoe & Hahn (2017), or questionnaire design Bellio et al. (2020). In neuropsychological evaluation for example, multiple questions have highly correlated answers and a key challenge is to identify multiple small combinations of questions that can evaluate e.g. dementia patients quickly and variably to avoid practice effects. Small dense data sets are also achievable in other ED scenarios, although the strategy may not extend to all. Future work will study the size and diversity of densely sampled data needed to ensure strong generalizable design.

Unlike alternative approaches in supervised FS e.g. Lee et al. (2022); Wojtas & Chen (2020), JOFSTO does not purport to consistently identify a small, highly discriminative feature subset Kuncheva et al. (2020); it is not designed to ignore a large number of nuisance features masking the variability of structured features Lindenbaum et al. (2021) or to explicitly integrate the correlation structure between feature subsets for efficiency Lee et al. (2022). This is an advantage in the ED paradigm, but means that JOFSTO may perform less well supervised FS tasks with these requirements, e.g. classification on some (but not all) proteins/DNA/RNA datasets, or identifying structured features of the synthetic noisy two-moon dataset. As JOFSTO uses DL, it is likely to overfit when the number of samples $n$ is limited, or $n$ is less than the number of selected features $C$ Kuncheva et al. (2020); Wojtas & Chen (2020). This scenario is frequent in e.g. genomics and could potentially occur even in imaging tasks if the diversity of signals over the image is low or the image is small e.g. in Ferizi et al. (2017). JOFSTO's use of RFE is to aid the challenging optimization by reducing to stepwise consideration of nested feature subsets across steps $t = 1, .., T$. However, this decreases the upper bound on performance as the optimal feature set for $C_t$ may not be a subset of the optimal set for $C_{t-1}$; future work will consider alternative optimization strategies to seek globally optimal designs.

### Ethics Statement

This paper highlights direct impact of our novel approach to ED applied to MRI and hyperspectral imaging. By selecting few acquisition parameters which may be obtained in routine (human-obtained) MRI scans, and performing a task that provides improved visual or microstructural information, a clinician may better analyze and diagnose a variety of pathologies, potentially improving public health. This of course depends on the specific acquisition platforms and site, which varies scan

performance. Thus results should be thoroughly verified by clinical experts before deployment. This paper also used in-vivo human brain MRI scans and followed the respective guidelines, where the authors of the cited, respective papers completed appropriate ethical reviews.

REPRODUCIBILITY

The code for our method in this paper will be publicly available, and is submitted to the reviewers Code (2022). All data are publicly available. Data acquired from MRI are described in section F, including processing, splits, and download links. Code to generate simulated data is in the code repository. Settings for the JOFSTO and baseline methods are in section B. Details on MRI model fitting is in section D and uses publicly available libraries.

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

# A    JOFSTO'S OPTIMIZATION ALGORITHM

This section presents further details concerning JOFSTO's optimization algorithm, outlined in section 3. We first describe the computational graph, illustrated in figure 1. We then describe the optimization procedure in full, described in algorithm 2. For steps $t = 1, ..., T$ this procedure obtains $m_t, \bar{s}_t, \mathcal{T}_t$, which are $m, \bar{s}, \mathcal{T}$ at the completion of step $t$. Multiplication follows standard array broadcasting rules e.g. $\mathbf{z}_1 \in \mathbb{R}^d, Z_2 \in \mathbb{R}^{n \times d}$, then $\mathbf{z}_1 \cdot Z_2 \in \mathbb{R}^{n \times d}$ is element-wise multiplication broadcasted to the first dimension.

**Computational Graph** This describes JOFSTO's computational graph during training, illustrated in figure 1-top and described in algorithm 1. JOFSTO has two neural networks: a Scoring Network $\mathcal{S}$ and a Task Network $\mathcal{T}$. The final feature score $s$ is a linear combination of a learnt score $s^p = \sigma(\mathcal{S}(\bar{X}))$ and a sample-independent (averaged) score $\bar{s}$, expressed as $s = \alpha^s \cdot s^p + (1 - \alpha^s) \cdot \bar{s}$, where $s^p \in \mathbb{R}^{n \times \bar{C}}, \bar{s} \in \mathbb{R}^{\bar{C}}$, and $\alpha^s \in [0, 1]$ is not learnt but used to progressively set $s = \bar{s}$, and $\sigma : \mathbb{R} \to [0, \infty)$ is an activation function to ensure positive scores. JOFSTO uses a mask $m \in [0, 1]^{\bar{C}}$ to subsample the measurements with $X = m \cdot \bar{X} + (1 - m) \cdot \bar{X}^{fill}$ where $\bar{X}^{fill} \in \mathbb{R}^{\bar{C}}$ is user-chosen (e.g. zeros or data median) and fills the locations of un-subsampled measurements. The second neural network, the Task Network $\mathcal{T}$ takes the weighted, subsampled measurements and estimates the target $\widehat{Y} = \mathcal{T}(s \cdot X)$, where multiplication here allows gradients to flow end-to-end. A supervised loss is calculated $L(\widehat{Y}, Y)$, for user-chosen $L$, allowing computation of $\mathcal{S}, \mathcal{T}$ gradients.

**Recursive Feature Elimination (RFE)** The user prespecifies the number of features to consider across steps $t = 1, ..., T$: $C_1, ..., C_T$ where $\bar{C} = C_1 > ... > C_T$, so $C_{t-1} - C_t$ features are removed in step $t > 2$. In step $t$ JOFSTO uses its novel approach to score features with $\bar{s}_t$, and uses $m_t$ to subsample the features, within a RFE framework.

**Step t = 1** On the first step JOFSTO trains on full information and learns a sample-independent score. It sets $m = m_1 = [1]^{\bar{C}}, \alpha^s = 1$ and trains across epochs $e = 1, ..., E$. At completion, set the first sample-independent score $\bar{s}_1 = \bar{s}^p$ where $\bar{s}^p$ is the mean of $s^p$ across samples/batches.

**Steps t = 2,...,T** For subsequent steps, JOFSTO splits the inner loop training procedure $1 <= E_1 < E_2 < E_3 < E$ for hyperparameters epochs $E_1, E_2, E_3$.

1. Across $e = 1, ..., E_1$ JOFSTO learns a sample-independent score for each measurement. It trains on a combination of (previous) sample-independent score and current learnt score, and at $e = E_1$ it sets the sample-independent score as a the mean of learnt scores. This is $\bar{s} = \bar{s}_{t-1}, \alpha^s = \frac{1}{2}, m = m_{t-1}$, at $e = E_1$ set $\bar{s}_t = \frac{1}{2}(\bar{s}_{t-1} + \bar{s}^p)$ where $\bar{s}^p$ is the mean of $s^p$ across samples/batches.

2. Across $e = E_1 + 1, ..., E_2$ JOFSTO progressively sets the score to be sample-independent, by progressively setting $\alpha^s$ from 1 to 0 successively on these epochs, allowing training to go from sample-dependent ($s = s^p$) to sample-independent scores ($s = \bar{s}$).

3. Across $e = E_2 + 1, ..., E_3$ JOFSTO chooses the lowest-scored measurements to subsample, then progressively modifies mask elements to perform this subsampling. At epoch $e = E_2$ choose the measurements to subsample $D_t = \arg \min_{C_{t-1} - C_t} \{\bar{s}_t[i] : m_{t-1}[i] = 1\}$, with mask target $m_t = m_{t-1} - \mathbb{I}_{i \in D}$. Construct the mask progressively with $m = \alpha^m \cdot m_{t-1} + (1 - \alpha^m) \cdot m_t$ where $\alpha^m \in [0, 1]$ is set from 1 to 0 across these epochs.

4. Across $e = E_3 + 1, ..., E$ the Task Network $\mathcal{T}$ is trained. At the completion of step $t$, this process returns (trained) $\mathcal{T}$ and $m, \bar{s}$, denoted as $\mathcal{T}_t, m_t, \bar{s}_t$. Then use as $\widehat{Y} = \mathcal{T}_t(X \cdot \bar{s}_t)$ as figure 1 Test.

**Using JOFSTO** For $C_t$ acquisition-parameter combinations, use $A^{C_t} = \{\mathbf{a}^c : m_t[c] = 1, c = 1, ..., \bar{C}\}$ to obtain new data, where $\mathbf{a}^c \in A^{C_t}$ produces the measurement $x_1^c, ..., x_n^c$. Then one fills un-acquired measurements with $\bar{X}^{fill}$ by setting $X_i^c = (x_i^c$ if $m[c] = 1$, else $\bar{X}^{fill}[c])$ producing $X \in \mathbb{R}^{n \times \bar{C}}$. Then estimate $Y$ with $\widehat{Y} = \mathcal{T}_t(X \cdot \bar{s}_t)$, and compute a downstream metric on $\widehat{Y}$.

**Implementation** JOFSTO's code is in PyTorch Paszke et al. (2019) and is submitted anonymously Code (2022). We describe its hyperparameters in section B.

---

**Algorithm 1** JOFSTO Forward and Backwards Pass: FPBP()

---

**Input** Input Data $\bar{X} \in \mathbb{R}^{n \times \bar{C}}$, Target Data $Y$, Subsampled Data Fill $\bar{X}^{fill}$, Scoring Network $\mathcal{S}$, Task Network $\mathcal{T}$, Final Feature Score Multiplier $\alpha^s \in [0,1]$, Mask $m \in [0,1]^{\bar{C}}$

1: $s^p = \sigma(\mathcal{S}(\bar{X}))$           ▷ Learn Sample-Dependent Feature Score
2: $s = \alpha^s \cdot s^p + (1 - \alpha^s) \cdot \bar{s}$          ▷ Final Feature Score
3: $X = m \cdot \bar{X} + (1 - m) \cdot \bar{X}^{fill}$      ▷ Measurement Subsampling with a Mask
4: $\widehat{Y} = \mathcal{T}(s \cdot X)$              ▷ Perform the Task
5: $L = L(\widehat{Y}, Y)$             ▷ Calculate Supervised Loss
6: Backpropagate on $L$, compute gradients of $\mathcal{S}, \mathcal{T}$ weights, update weights of $\mathcal{S}, \mathcal{T}$

---

**Algorithm 2** JOFSTO Optimization

---

**Input** Input Data $\bar{X} \in \mathbb{R}^{n \times \bar{C}}$, Target Data $Y$, Subsampled Data Fill $\bar{X}^{fill}$, Feature Size Candidates and RFE Steps $\bar{C} = C_1 > ... > C_T$, Training Steps $1 <= E_1 < E_2 < E_3 < E$

1: **for** $t \leftarrow 1$ **do**
2:   $m \leftarrow m_1 \leftarrow [1]^{\bar{C}}, a^s \leftarrow 1$
3:   **for** $e \leftarrow 1, ..., E$ **do**          ▷ Train on Full Information
4:    FPBP()
5:   **end for**
6:   Calculate $\bar{s}^p$ the mean learnt score of $s^p$ across data
7:   $\bar{s}_1 \leftarrow \bar{s}^p$           ▷ First Sample-Independent Score
8: **end for**
9: **for** $t \leftarrow 2, ..., T$ **do**
10:   $\bar{s} \leftarrow \bar{s}_{t-1}, \alpha^s \leftarrow \frac{1}{2}, m \leftarrow m_{t-1}$
11:   **for** $e \leftarrow 1, ..., E_1$ **do**         ▷ Joint Optimization
12:    FPBP()
13:   **end for**
14:   Calculate $\bar{s}^p$ the mean learnt score of $s^p$ across data
15:   $\bar{s}_t \leftarrow \frac{1}{2}(\bar{s}_{t-1} + \bar{s}^p)$       ▷ Sample-Independent Score for step $t$
16:   **for** $e \leftarrow E_1 + 1, ..., E_2$ **do**
17:    $\alpha^s \leftarrow \max\{\alpha^s - \frac{2}{E_2 - E_1}, 0\}$   ▷ Progressively Set the Score to be Sample-Independent
18:    FPBP()
19:   **end for**
20:   $D_t \leftarrow \arg\min_{C_{t-1} - C_t}\{\bar{s}_t[i] : m_{t-1}[i] = 1\}$     ▷ Features to Subsample
21:   $m_t \leftarrow m_{t-1} - \mathbb{I}_{i \in D}$          ▷ Mask Target
22:   $\alpha^m \leftarrow 1$
23:   **for** $e \leftarrow E_2 + 1, ..., E_3$ **do**
24:    $\alpha^m \leftarrow \max\{\alpha^m - \frac{1}{E_3 - E_2}, 0\}$
25:    $m \leftarrow \alpha^m \cdot m_{t-1} + (1 - \alpha^m) \cdot m_t$    ▷ Progressively Construct the Mask
26:    FPBP()
27:   **end for**
28:   **for** $e \leftarrow E_3 + 1, ..., E$ **do**
29:    FPBP()
30:   **end for**
31:   Cache $T_t, m_t, \bar{s}_t$, then use as $\widehat{Y} = \mathcal{T}_t(X \cdot \bar{s}_t)$     ▷ See figure 1-Test
32: **end for**

---

# B    SETTINGS FOR DIFFERENT APPROACHES

This section describes the settings for the different approaches used in this paper.

We used the official code for all baselines when required, following the implementation carefully. For every approach, for fairness, we considered the same number of evaluations on the validation data, to choose the best model to apply to the test data, for each different feature set size $C$. Data are split in training, validation and test sets, described in section F. We normalize the data following Grussu et al. (2021), which divides each channel/measurement by its $99th$ percentile value calculated from the training set, the input and output of the neural network are normalized by this value. Other general hyperparameters are batch size 1500, learning rate $10^{-4}$ ($10^{-5}$ in section 4.1.1) and $ADAM$ optimizer, default network weight initialization.

## OURS: JOFSTO

We conducted a brief search for the hyperparameters, setting $E_1 = 25, E_2 = E_1 + 10, E_3 = E_2 + 10$, we do not set $E$, as like other approaches, we train on $e = E_3, ....$ until early stopping on the validation set. We linearly modify the values of $\alpha^s$ in epochs $e = E_1 + 1, ..., E_2$ and linearly modify $m$ in epochs $e = E_2 + 1, ..., E_3$. The activation function for the score is $\sigma = 2 \times sigmoid$, so the feature scores are initialized to approximately 1. The subsampled data fill $\bar{X}^{fill} \in \mathbb{R}^{\bar{C}}$ is the median of the data across each measurement/channel. These hyperparameters were constant across all experiments. For each experiment we chose $C_i$ to be the subset sizes required for the task e.g. in table 1 we set $C_{1,2,3,4,5} = \bar{C}, \frac{\bar{C}}{2}, \frac{\bar{C}}{4}, \frac{\bar{C}}{8}, \frac{\bar{C}}{16}$.

We perform a grid search to find the optimal network architecture hyperparameters for each task. The Scoring Network $\mathcal{S}$ and Task Network $\mathcal{T}$ have the same number of hidden layers $\in \{1, 2, 3\}$, number of units $\in \{30, 100, 300, 1000, 3000\}$, for each combination we obtain task performance on the feature set sizes $C_1 > C_2 > ....$ The best performing network on the validation set is deployed on the test data.

## RANDOM FEATURE SELECTION (FS) + DEEP LEARNING (DL)

For different $C$ we repeat the following process. We randomly select $C$ number of measurements on a fixed random seed and perform grid search on the task network (mapping subsampled data to full data), with number of hidden layers $\in \{1, 2, 3\}$, number of units $\in \{30, 100, 300, 1000, 3000\}$. Training is until early stopping on the validation set. The best trained model is evaluated on the test set.

## LEE ET AL. (2022)

This approach has a three-stage procedure and four neural networks to exploit task-performance, self-supervision, additional unlabeled data, and correlated feature subsets. It scores the features then subsequently train a task-based network on subsampled data. We use the official repository Lee (2022). The optimization procedure is split into i) self-supervision phase, ii) supervision phase, iii) training on selected features only.

The self-supervision phase finds the optimal encoder network hyperparameters. We perform grid search, the encoder network, feature vector estimator network, gate vector estimator network, all have same number of hidden layers $\in \{1, 2, 3\}$ number of units, including hidden dimension $\in \{30, 100, 300, 1000, 3000\}$, following Lee et al. (2022)-table-S.1, other hyperparameters $\alpha \in \{0.01, 0.1, 1.0, 10, 100\}, \pi \in \{0.2, 0.4, 0.6, 0.8\}$. The self-supervisory dataset is training set $X$. On the best validation performance (with early stopping), this returns a trained encoder network, cached for the supervision phase.

The supervision phase scores the features. The pre-trained encoder is loaded from the previous phase. We then perform grid search, where the predictor network has number of hidden layers $\in \{1, 2, 3\}$, number of units $\in \{30, 100, 300, 1000, 3000\}$, following table S.1 $\beta \in \{0.01, 0.1, 1.0, 10, 100\}$. On the best validation performance with early stopping, the process returns a score for all features.

The final phase is repeated for different number of features subset sizes $C$. We extract the $C$ highest scored features from the previous phase and perform grid search on the task network (mapping subsampled data to full data), with number of hidden layers $\in \{1, 2, 3\}$, number of units $\in \{30, 100, 300, 1000, 3000\}$. Training is until early stopping on the validation set. The best trained model is evaluated on the test set.

WOJTAS & CHEN (2020)

This approach has a complicated three-stage procedure and two neural networks that respectively performs the task, and scores masks, where the two networks are trained alternatively. This returns an optimal mask (analogous to $m_t$ in our approach) and trained task-network (analogous to $\mathcal{T}_t$ in our approach). We use the official repository Wojtas (2021). The following process is repeated for different feature subset sizes $C$.

We perform a grid search to find the optimal hyperparameters. The operator network (analogous to task network $\mathcal{T}$ in this paper) and the selector network have the same number of hidden layers $\in \{1, 2, 3\}$, number of units $\in \{30, 100, 300, 1000, 3000\}$, $s_p = 5, E_1 = 15000$. The joint training uses early stopping on the validation set, and returns an optimal feature set size, of size $C$ and a trained operator network. The best performing operator network on the validation set is deployed on the test data.

## C   THE NODDI AND VERDICT MRI MODELS

This section describes the VERDICT and NODDI models used in section 4.1.1.

1. The VERDICT (Vascular, Extracellular and Restricted Diffusion for Cytometry in Tumors) model Panagiotaki et al. (2014), which maps histological features of solid-cancer tumors particularly for early detection and classification of prostate cancer Panagiotaki et al. (2015a); Johnston et al. (2019); Singh et al. (2022). The VERDICT model includes parameters: $f_I$ the intra-cellular volume fraction, $f_V$ the vascular volume fraction, $D_v$ the vascular perpendicular diffusivity, $R$ the mean cell radius, and **n** - a 3D vector defining mean local vascular orientation.

2. The NODDI (Neurite Orientation Dispersion and Density Imaging) model Zhang et al. (2012), maps parameters of the cellular composition of brain tissue and is widely used in neuroimaging studies in neuroscience such as the UK Biobank study Alfaro-Almagro et al. (2018), and neurology e.g. in Alzheimer's disease Kamiya et al. (2020) and multiple sclerosis Grussu et al. (2017). The NODDI model includes five tissue parameters: $f_{ic}$ the intra-cellular volume fraction, $f_{iso}$ the isotropic volume fraction, the *orientation dispersion index* (ODI) that reflects the level of variation in neurite orientation, and **n** - a 3D vector defining the mean local fibre orientation.

Implementations of both models are publicly available in Fick et al. (2019).

## D   HOW TO FIT DTI, DKI AND MSDKI MODELS AND COMPUTE DOWNSTREAM METRICS (PARAMETER MAPS)

This section provides additional information for the DTI, DKI and MSDKI models, how they are fitted on MRI data, and how to compute downstream metrics, used in section 4.1.3.

Diffusion tensor imaging (DTI) Basser et al. (1994), diffusion kurtosis imaging (DKI) Jensen & Helpern (2010), and Mean Signal DKI (MSDKI) are widely-used qMRI methods. Like NODDI and VERDICT, they use diffusion MRI to sensitise the image intensity to the Brownian motion of water molecules within the tissue to provide a window on tissue microstructure. However, whereas NODDI and VERDICT are designed specifically for application to brain tissue and cancer tumours, respectively, DTI and DKI are more general purpose techniques that provide indices of diffusivity (e.g. mean diffusivity - MD), diffusion anisotropy (e.g. fractional anisotropy Basser & Pierpaoli (1996) - FA), and the deviation from Gaussianity, or kurtosis, (e.g. mean kurtosis Jensen & Helpern (2010) - MK) that can inform on tissue integrity or pathology. Mean signal diffusion kurtosis imaging (MSDKI) is a simplified version of DKI that quantifies kurtosis using a simpler model that is easier to fit Henriques (2018). These techniques show promise for extracting imaging biomarkers for a wide

Table 8: Mean-squared-error between parametric maps (downstream metrics) estimated from $\bar{C} = 288$ ground truth measurements and $\tilde{C}$ from $C$ reconstructed measurements. Additional results to table 5.

| $C = 36$ | DTI | | | | DKI | | | MSDKI | |
|---|---|---|---|---|---|---|---|---|---|
| | FA | MD | AD | RD | MK | AK | RK | MSD | MSK |
| Random FS + DL | 1.17 | 3.13 | 9.3 | 3.75 | 6.37 | 6.28 | 13.3 | **2.50** | 4.15 |
| Lee et al. (2022) | 5.96 | 10.4 | 43.7 | 14.2 | 8.74 | 7.27 | 16.9 | 3.87 | 10.8 |
| Wojtas & Chen (2020) | 10.4 | 68.8 | 106 | 81.4 | 9.39 | 10.3 | 18.6 | 11.1 | 8.82 |
| JOFSTO | **0.94** | **1.73** | **8.15** | **1.50** | **6.15** | **5.92** | **12.6** | 2.75 | **3.78** |

variety of medical applications, such mild brain trauma, epilepsy, stroke, and Alzheimer's disease Ranzenberger & Snyder (2022); Tae et al. (2018); Jensen & Helpern (2010).

This section provides an outline of the standard MRI model fitting approaches we used in section 4.1.3. To fit the DTI,DKI,MSDKI MRI models to the data, and obtain the downstream metrics (parameter maps), we employ the widely-used, open-source DIPY library Garyfallidis et al. (2014). We followed standard practice for model fitting in MRI and used the least-squares optimization approach and default fitting settings. To remove outliers, values were clamped where DTI FA $\in [0, 1]$, DTI MD,AD,RD $\in [0, 0.003]$, DKI MK,AK,RK $\in [0, 3]$, MSDKI MSD $\in [0, 0.003]$ MSDKI MSK $\in [0, 3]$.

# E    ADDITIONAL RESULTS ON ESTIMATING TISSUE MICROSTRUCTURE FROM WATER DIFFUSIVITY

This section provides additional results to section 4.1.3 in table 8

# F    DATA

This section describes the data used in this paper. We provide a summary in table 9. Visualizations of slices are in figure 4.

Table 9: Summary of data.

| Name | Type | No. Measurements $\bar{C}$ | Resolution $mm^3$ | Acquisition Parameter Dimension | No. voxels $n$ (K) Train | Val | Test |
|---|---|---|---|---|---|---|---|
| MUDI | qMRI scan | 1344 | 2.5 | $A \subset \mathbb{R}^6$ | 321 | 132 | 105 |
| HCP | qMRI can | 288 | 1.25 | $A \subset \mathbb{R}^4$ | 2182 | 774 | 674 |
| VERDICT | Simulated qMRI | 220 | - | $A \subset \mathbb{R}^7$ | 1000 | 100 | 100 |
| NODDI-WMC | Simulated qMRI | 3612 | - | $A \subset \mathbb{R}^7$ | 100 | 10 | 10 |

VERDICT AND NODDI MRI MODELS SIMULATED DATA

Table 10: Parameter ranges for simulating synthetic VERDICT and NODDI model data,

| VERDICT | | | NODDI | | |
|---|---|---|---|---|---|
| Parameter | Minimum | Maximum | Parameter | Minimum | Maximum |
| $f_I$ | 0.01 | 0.99 | $f_{ic}$ | 0.01 | 0.99 |
| $f_V$ | 0.01 | 0.99 | $f_{iso}$ | 0.01 | 0.99 |
| $D_v$ ($\mu$ms$^2$ s$^{-1}$) | 3.05 | 10 | $ODI$ | 0.01 | 0.99 |
| $R$ ($\mu$m) | 0.01 | 20 | $\mathbf{n}$ | [-1 -1 -1] | [1 1 1] |
| $\mathbf{n}$ | [-1 -1 -1] | [1 1 1] | | | |

To simulate synthetic data for the VERDICT and NODDI models we use a known, fixed, acquisition scheme in combination with a set of ground truth model parameters. We choose the ground truth model parameters by uniformly sampling parameter combinations from the bounds given in table 10. We choose these bounds as they approximate the physically feasible limits of the parameters.

The VERDICT data has number of samples $n = 1000K, 100K, 100K$ in the train, validation, test split, with target data $Y \in \mathbb{R}^{n \times 8}, \boldsymbol{\theta}_i \in \mathbb{R}^8, i = 1, ..., n$. The classical ED approach yields an acquisition scheme derived from the Fisher information matrix Panagiotaki et al. (2015b) and here $X \in \mathbb{R}^{n \times 20}, C = 20$. The approaches in supervised feature selection use a densely-sampled empirical acquisition scheme, designed specifically for the VERDICT protocol from Panagiotaki et al. (2015a) and here $\bar{X} \in \mathbb{R}^{n \times 220}, \bar{C} = 220$.

The NODDI data has number of samples $n = 100K, 10K, 10K$ in the train,validation,test split, with target data $Y \in \mathbb{R}^{n \times 7}, \boldsymbol{\theta}_i \in \mathbb{R}^7, i = 1, ..., n$. The classical ED approach yields an acquisition scheme derived from the Fisher information matrix Zhang et al. (2012) and so $X \in \mathbb{R}^{n \times 99}, C = 99$. The approaches in supervised feature selection use a densely-sampled empirical acquisition scheme from an extremely rich acquisition from Ferizi et al. (2017). This was designed for the ISBI 2015 White Matter Challenge, which aimed to collect the richest possible data to rank MRI models, and required a single subject to remain motionless for two uncomfortable back-to-back 4h scans. Here $\bar{X} \in \mathbb{R}^{n \times 3612}, \bar{C} = 3612$.

We used Rician noise, as appropriate for MRI data Gudbjartsson & Patz (1995). The signal to noise ratio of the unweighted signal is 50, which is representative of clinical qMRI.

To conduct the simulations we employ the widely-used, open-source dmipy toolbox Fick et al. (2019), and the code for the simulations is anonymously submitted. The results in table 5 are all scaled by: DTI-FA $\times 10^2$, DTI-MD $\times 10^9$, DTI-AD $\times 10^9$, DTI-RD $\times 10^9$, DKI-MK $\times 10^2$, DKI-AK $\times 10^2$, DKI-RK $10^2$, MSDKI-MSD $\times 10^9$, MSDKI-MSK $\times 10^2$.

### F.1 MULTI-DIFFUSION (MUDI) CHALLENGE DATA

Data used in section 4.1.2 are images from 5 in-vivo human subjects, and are publicly available MUDI Organizers (2022), and was acquired with the state-of-the-art ZEBRA sequence Hutter et al. (2018). Diffusion-relaxation MRI has a 6D acquisition parameter space $A \subset \mathbb{R}^6$: echo time (TE), inversion time (TI), b-value, and b-vector directions in 3 dimensions: $b_x, b_y, b_z$. Data has $2.5mm$ isotropic resolution and field-of-view $220 \times 230 \times 140mm$ and resulted in 5 3D bran images with $\bar{C} = 1344$ measurements/channels, which here are unique diffusion- $T2^*$ and $T1$- weighting contrasts. More information is in Hutter et al. (2018); Pizzolato et al. (2020). Each subject has an associated brain mask, after removing outlier voxels resulted in $104520, 110420, 105743, 132470, 105045$ voxels for respective subjects $11, 12, 13, 14, 15$. For the experiment in table 4, we followed the original MUDI challenge Pizzolato et al. (2020) and took subjects $11, 12, 13$ as the training and validation set, and subjects $14, 15$ as the unseen test set, where $90\% - 10\%$ of the training/validation set voxels are respectively, for training and validation.

### F.2 HUMAN CONNECTOME PROJECT (HCP) TEST RETEST DATA

In section 4.1.3 we utilise WU-Minn Human Connectome Project (HCP) diffusion data, which is publicly available at `humanconnectome.org` (Test Retest Data Release, release date: Mar 01, 2017) Essen et al. (2013). The data comprises $\bar{C} = 288$ volumes (i.e. measurements/channels), with 18 $b = 0$ (i.e. non-diffusion weighted) volumes, 90 gradient directions for $b = 1000$ s mm$^{-2}$, 90 directions for b=2000 s mm$^{-2}$, and 90 directions for b=3000 s mm$^{-2}$. We used 3 scans for training $103818\_1, 105923\_1, 111312\_1$, one scan for validation $114823\_1$ and one scan for testing $115320\_1$, which produced number of samples $n = 708724 + 791369 + 681650 = 2181743, 774149, 674404$ for the respective splits. We only used voxels inside the provided brain mask and normalized the data voxelwise with a standard technique in MRI, by dividing all measurements by the mean signal in each voxel's $b = 0$ values. Undefined voxels were then removed.

Figure 3: Correlation coefficient between the measurements/features of the data, zero entries correspond to b0-values.

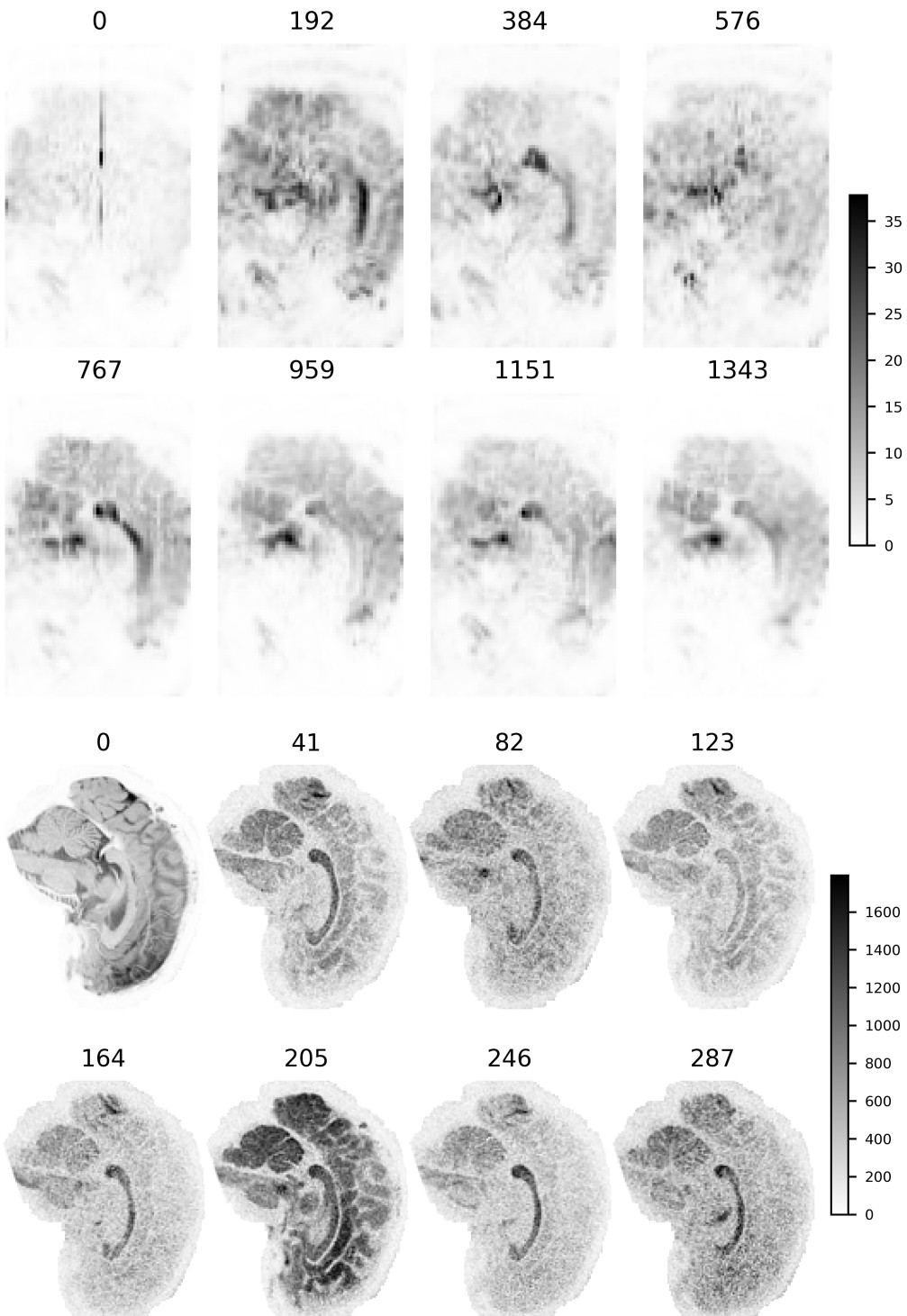

Figure 4: 2D brain slices from 3D MRI scans, for different measurements/features/channels for the MUDI (above) HCP (below) data.

Table 11: Ablation study on JOFSTO's components, experimental settings in table 1.

| $C =$ | 110 | 55 | 28 | 14 |
|---|---|---|---|---|
| JOFSTO w/o Scoring Network $S$ | 7.23 | 10.7 | 11.5 | 11.5 |
| JOFSTO w/o RFE | **1.03** | 1.19 | 1.83 | 2.80 |
| JOFSTO | **1.03** | **1.18** | **1.80** | **2.64** |

Table 12: Standard deviation of performance $\times 10^2$ and mean Jaccard Index between chosen measurements, across 10 random seeds, experimental settings in table 1 VERDICT simulations.

| | STD performance across 10 seeds | | | | Mean Jaccard of Chosen Measurements % | | | |
|---|---|---|---|---|---|---|---|---|
| $C =$ | 500 | 250 | 100 | 50 | 500 | 250 | 100 | 50 |
| Random FS + DL | 0.11 | 0.23 | 0.37 | 0.76 | 32.8 | 15.2 | 6.82 | 2.87 |
| Lee et al. (2022) | 0.01 | 0.02 | 0.05 | 0.18 | 81.2 | 71.4 | 62.2 | 75.9 |
| Wojtas & Chen (2020) | 0.44 | 0.34 | 0.37 | 0.44 | 34.3 | 18.1 | 48.8 | 41.0 |
| JOFSTO | 0.01 | 0.02 | 0.01 | 0.12 | 74.3 | 82.1 | 84.6 | 59.1 |

## F.3 ABLATION STUDY AND THE EFFECT ON RANDOMNESS ON PERFORMANCE

Table 11 examines the impact of removing JOFSTO's components on performance. First it considers JOFSTO without RFE i.e. without iteratively removing features in our optimization algorithm, fixing $t = 2$ and $C_1, C_2 = \bar{C}, C$, showing that iterative subsampling has better performance than subsampling all the features in a single iteration. As our approach to feature scoring is a key element of JOSFTO, we also consider the impact of removing the scoring network $S$, which results in extremely poor performance as training is destabilized when progressively setting the score $s$ from sample-dependence to sample-independence. Table 12 examines how the random seed affects network initialization and data shuffling, impacts performance. Results show JOFSTO performs favourably compared to alternative approaches, and JOFSTO is mostly robust to randomness inherent in DL.

