# OpenReview forum: "An Experiment Design Paradigm using Joint Feature Selection and Task Optimization"
_ICLR.cc/2023/Conference — Submitted to ICLR 2023_

### Official Review · Reviewer_SJC9 · 2022-10-24

**Confidence:** 3
**Correctness:** 3
**Technical Novelty And Significance:** 2
**Empirical Novelty And Significance:** 3
**Recommendation:** 3

**Clarity, Quality, Novelty And Reproducibility:**

The paper presents a practical solution for designing qMRI, but unfortunately I do not see significant contributions for the AI literature in general. While the basic idea of simultaneously optimizing for the features and the experimental design is reasonable, the actual solution requires access to data that corresponds to essentially having already solved the problem. The authors explain well why this data can be obtained in the specific case of qMRI and the work is hence clearly publishable, but I do not think ICLR is the right venue for such a specialised application. The bare minimum would be to explain a few other examples where obtaining a super-design that densely covers the samples together with target labels that correspond to the same task is realistic and the actual goal is to obtain an economical design, and then re-write the paper using a more general terminology. I can think of some possible examples at least in inverse modelling, but the authors should be much more detailed in justifying the relevance of the problem formulation in general.

The other main problem is that the method description is very shallow. Section 3 takes less than a page, the paper has no formal description of the model but only explains it in Figure 1 (which cannot be understood in detail as some symbols are not even defined), and there are no justifications for any of the choices; the algorithm is simply explained to consist of an outer an inner loop, with no discussion on why it has to be done like this. In technical terms, the solution looks like a specific neural network architecture with fairly standard training algorithm.

The empirical experiments are good, but quite specialised for the application domain. They certainly warrant publishing the work in a suitable venue.

**Strength And Weaknesses:**

Strengths:
- The approach is very well motivated for the qMRI application where the effect of different acquisition parameters is reflected in a different manner in each voxel
- Works well in several experiments

Weaknesses:
- The method explicitly relies on densely sampled super-design and access to ground truth labels for 'the task driving the ED'. While the authors explain why we can satisfy these assumptions in the specific case of qMRI, I cannot identify other common scenarios where we would meet these criteria and the goal would be to find an 'economical design' for the same task. This severely limits the value of the work for general AI audiences
- The paper is presented strongly from the perspective of the application (as it needs to be because the details rely on the assumptions that hold for this application) and would better fit a more specialised venue.

**Summary Of The Paper:**

The authors propose an approach for experiment design in context of qMRI studies. The new approach combines feature selection and the choice of tasks in an integrated solution, using a combination of two neural networks.

**Summary Of The Review:**

A novel and potentially interesting model for setting the configuration of qMRI devices, but the paper does not feel like a good fit for ICLR since the method makes assumptions that are not easy to satisfy in general cases and the method presentation is not detailed enough. I cannot see future work building on this.

---

> ### Author Response · Authors · 2022-11-19
> **Response to Reviewer SJC9**
>
> Thank you for your feedback.
>
> > cannot identify other common scenarios where we would meet these criteria and the goal would be to find an 'economical design \
> few other examples where obtaining a super-design that densely covers the samples together with target labels that correspond to the same task is realistic \
> Bare minimum ... explain a few other examples where obtaining a super-design that densely covers the samples together with target labels that correspond to the same task is realistic and the actual goal is to obtain an economical design ... re-write the paper using a more general terminology
>
> We appreciate the reviewer's view, since we only included results from one application area, but the proposed method has wider applicability, as it makes no assumption about the data and may be applied to other problems in supervised feature selection.  We have now included two additional results on a distinct application: hyperspectral imaging in section 4.2, and note that hyperspectral imaging, in general, has many applications.  In the discussion section 5, we elaborate on how our results on MRI and hyperspectral imaging point to additional potential applications.  We also discuss additional application areas, specifically neuropsychological questionnaire design, and also point to cell populations.
>
> > better fit a more specialised venue. \
> do not think ICLR is the right venue for such a specialised application \
> not see significant contributions for the AI literature in general. \
> empirical experiments are good, but quite specialised for the application domain. ... warrant publishing the work in a suitable venue
>
> Even if one were to ignore our new results in section 4.2, we strongly disagree that ICLR is an incorrect venue for the paper as:
>
> The conference remit includes 'applications in audio, speech, robotics, neuroscience,  biology, or any other field' [link](https://iclr.cc/Conferences/2023), and `Please Choose The Closest Area That Your Submission Falls Into: Machine Learning for Sciences (eg biology, physics, health sciences, social sciences, climate/sustainability )', so even focussed on qMRI we believe the work is within remit
>
> The paper has wider interest, as it highlights new challenges within feature selection - a core topic for ICLR - and, as we now demonstrate, the ED paradigm has wider applicability i.e. 'formulated within the Experimental Design setting, but is also correctly positioned within the feature selection literature' (Reviewer SWxw)
>
> ICLR has accepted MRI-focused papers e.g. Zero-Shot Self-Supervised Learning for MRI Reconstruction (ICLR 2022).
>
> > method description is very shallow. Section 3 takes less than a page, the paper has no formal description of the model but only explains it in Figure 1 ... algorithm is simply explained to consist of an outer an inner loop, with no discussion on why it has to be done like this
>
> Due to lack of space, we could only provide high-level details in the main text. The full description of the method, referenced at the beginning of that section, is in the supplementaries section A (pg 14), which the paper points to.  We have moved parts of the supplementaries into the main text and enhanced the cross referencing to highlight the extent of the method development more clearly.  We have also endeavoured to highlight the novelty more clearly and motivate the choices for the problem we consider; see the new section 3.
>
> > technical terms, the solution looks like a specific neural network architecture with fairly standard training algorithm
>
> The key novelty lies in the new approach to a long-standing paradigm. The technical novelty over existing feature selection algorithms may be relatively simple, but is highly effective and essential for the novel application we enable. Nevertheless, we do feel that important technical innovations have been made. Our approach to perform concurrent feature scoring, feature subsampling, and task-prediction is novel. Our end-to-end differentiable dual-network that scores features and then performs a task is novel. Our specific optimization procedure that scores features to then progressively sample the features, and iteratively to drive the subsampling, is also novel. The novel features of our method is the reason for you saying 'empirical experiments are good', and 'Compelling performance against strong and recently published baselines' (Reviewer SWxw) on four datasets.
>
> > actual solution requires access to data that corresponds to essentially having already solved the problem.
>
> This is a misunderstanding. The JOFSTO paradigm requires densely-sampled (on the measurement space) data of $ \bar{C} $ measurements only for training, i.e. from a small number of subjects. From that it derives an economical protocol with much fewer measurements $ C << \bar{C} $  for long-term efficient use e.g. on patients. The problem is the identification of the small set of $ C $ measurements, which the dense acquisition does not solve.

---

### Official Review · Reviewer_SWxw · 2022-10-25

**Confidence:** 4
**Correctness:** 3
**Technical Novelty And Significance:** 2
**Empirical Novelty And Significance:** 2
**Recommendation:** 5

**Clarity, Quality, Novelty And Reproducibility:**

The presented approach seems to be in the spirit of the backward elimination wrapper approach to feature selection, which is well known for a quarter of century “Wrappers for feature subset selection” Kohavi & John (1997), so I would say that novelty is quite incremental. As mentioned, the paper is very well written, and results seems reproducible given that code will be shared and datasets are publicly available.

**Strength And Weaknesses:**

Strengths
Compelling performance against strong and recently published baselines.
The problem is formulated within the Experimental Design setting, but is also correctly positioned within the feature selection literature.
The writing style of the paper stands out, easily readable and comprehensible.

Weaknesses
It is unclear, at least to me, how the “selection of highly correlated globally informative (set of) candidates” is suitable to ED setting, while “small number of highly informative features among the many uninformative ones” is more suitable for FS. It all depends on the overall objective, and if both ED and FS share the same objective, they should favor the same/similar set of features. And how are those properties distinct/different?
While it is encouraging that JOFSTO is outperforming Lee at al. (2022) and Wojtas & Chen (2020) approaches, it is peculiar that random FS + DL is also outperforming them (Table 5.). Appears that somehow informed feature selection, during the network training, is actually hurting the performance. Can it be that some sort of overfitting is behind such results?
Ablation study (Table 6.) suggests that Recursive Feature Elimination is not bringing that much performance improvement. Is the RFE step necessary then?


**Summary Of The Paper:**

This paper presents a joint feature selection and model training procedure (JOFSTO) which is suitable for problems where features have certain spatial dependency properties (“densely sampled in a measurement space”). Procedure trains two connected neural networks, one which performs feature selection task, and the other which builds a prediction model on top of selected features. In particular application, the task is regression on quantitative MRI data, where JOFSTO had the smallest error against two recent state of the art joint feature selection and neural network training approaches.

**Summary Of The Review:**

Under the impression of limited novelty and some questions around approach motivation and empirical evaluation, I am leaning towards rejection, although very eager to hear arguments that would change my impressions.

---

> ### Author Response · Authors · 2022-11-18
> **Response to Reviewer SWxw**
>
> Thank you for the feedback.
>
> > in the spirit of the backward elimination wrapper approach to feature selection [BEWAFS], ... Kohavi & John (1997), so I would say that novelty is quite incremental
>
> Thank you for the reference, we acknowledged our approach is in a RFE framework, which itself could be viewed as being in the the BEWAFS framework and have added this to methods section 3.  As other feature selection papers (Wojtas et. al, Lee et. al), our novelty is in proposing a novel neural network architecture(s) and optimization procedure to perform feature selection, not in inventing an entirely novel feature selection paradigm.  More specifically, our novelty is (i) Performing concurrent feature scoring, feature subsampling, and task-prediction (ii) Proposing a novel end-to-end differentiable dual-network that scores features and then performs a task, (ii) An optimization algorithm to progressively remove features during training.  The novel features of our method is the reason for the 'Compelling performance against strong and recently published baselines' you say, on four datasets
>
> We also have conceptual innovations: recasting the experiment design (ED) setting as supervised feature selection and Results providing a proof-of-concept of our paradigm in the clinically-useful setting of qMRI, and in hyperspectral imaging, where we outperform classical approaches in ED, obtain state-of-the-art performance on two tasks and outperform state-of-the-art approaches in supervised feature selection on five datasets/tasks.
>
>
> >how the “selection of highly correlated globally informative (set of) candidates” is suitable to ED setting, while “small number of highly informative features among the many uninformative ones” is more suitable for FS.  And how are those properties distinct/different?
>
> We clarified pg1-2 to explain that state-of-the-art techniques in supervised feature selection are designed to search for a small number of highly informative features among the many uninformative ones.  However our problem domain has a different nature: the features in ED however are highly-correlated (figure 3) and globally informative.  Compare that to e.g. Lee et al. (2022) figure S.2.  This difference demands a rethink to the algorithm design, as we propose.
>
> > peculiar that random FS + DL is also outperforming them (Table 5.). Appears that somehow informed feature selection, during the network training, is actually hurting the performance. Can it be that some sort of overfitting is behind such results
>
> Overfitting is not why FS+DL outperforms Lee et al, as the neural networks deployed on the chosen features are the same size in each case (see section B).  We believe Lee et al's algorithm underperms on this problem, because of the highly informative and correlated set of features, which it is not designed to work with, so the features selected are no more informative than random selection.
>
> > Ablation study (Table 6.) suggests that Recursive Feature Elimination is not bringing that much performance improvement.
>
> Table 6 has been moved to Supplementaries-Table 11, and it shows that integrating JOFSTO's novel scoring and feature subsampling approach into a RFE framework improves performance by $ \approx 5.5 $ %  for the greatest subsampling rate $ C = 14 $.  This is the most important subsampling rate, requiring the least number of measurements to be acquired.

---

### Official Review · Reviewer_S5n8 · 2022-10-25

**Confidence:** 3
**Correctness:** 3
**Technical Novelty And Significance:** 2
**Empirical Novelty And Significance:** 3
**Recommendation:** 5

**Clarity, Quality, Novelty And Reproducibility:**

As mentioned previously, the clarity of the presentation needs to be improved.
Methodological novelty is somewhat limited, but the proposed scheme - JOFSTO - is shown to yield good consistent performance under a number of qMRI problems outperforming other baselines, demonstrating its potential practical advantage.



**Strength And Weaknesses:**

OVERALL COMMENTS

The main strength of the work lies in the overall performance gain achieved by JOFSTO in qMRI applications - in terms of improved parameter estimation, enhanced reconstruction performance, and more accurate quantification of tissue microstructure.
While the proposed method itself strongly builds on existing work - especially,  Wojtas & Chen (2020) that utilizes dual feature scoring and task prediction networks - and the underlying idea is relatively simple, the performance evaluation results show that it nevertheless has potential benefits in qMRI analysis.

However, the literary and technical presentations in the paper leave much room for improvement.
Several important details for understanding the proposed methodology are missing or unclear, and the authors often refer to examples without first explaining their relevance and some notations without providing a clear definition.

Furthermore, the proposed work appears to be motivated by a very specific application (i.e., ED for qMRI acquisition parameter optimization), and it is unclear whether JOFSTO would be applicable to other experiment design problems beyond qMRI applications considered in this study.


DETAILED COMMENTS

1. Considering that the proposed work is focused on a very specific application and problem setting, this should be more clearly reflected in the title and abstract.

2. Experimental design is a widely studied topic across various disciplines, while the authors seem to be mostly focused on "experiment design" in qMRI applications.
However, "ED in qMRI" seems to be quite different from what is typically referred to "experimental design" or "optimal experiment design (OED)" in various other fields, and this may potentially lead to confusion for readers.
It would be important to provide a more general treatment of the OED problem in the introduction or background and then zoom into this more specific ED problem in qMRI to avoid any confusion.

Here are a few relevant references:

Lindley, Dennis Victor. Bayesian statistics: A review. Society for industrial and applied mathematics, 1972.
Chaloner, Kathryn, and Isabella Verdinelli. "Bayesian experimental design: A review." Statistical Science (1995): 273-304.
Clyde, Merlise A. "Experimental design: A Bayesian perspective." International Encyclopia Social and Behavioral Sciences 8 (2001): 5075-5081.
Sebastiani, Paola, and Henry P. Wynn. "Maximum entropy sampling and optimal Bayesian experimental design." Journal of the Royal Statistical Society: Series B (Statistical Methodology) 62.1 (2000): 145-157.
Huan, Xun, and Youssef M. Marzouk. "Simulation-based optimal Bayesian experimental design for nonlinear systems." Journal of Computational Physics 232.1 (2013): 288-317.
Dehghannasiri, Roozbeh, Byung-Jun Yoon, and Edward R. Dougherty. "Optimal experimental design for gene regulatory networks in the presence of uncertainty." IEEE/ACM Transactions on Computational Biology and Bioinformatics 12.4 (2014): 938-950.
Foster, Adam, et al. "Variational Bayesian optimal experimental design." Advances in Neural Information Processing Systems 32 (2019).

3. The authors mention certain problems/applications - seemingly not closely related to the current work - without providing any context. Some examples include:
"e.g. selecting protein expressions for classification" (in the abstract)
"unlike for example protein-coding genes 10x Genomics (2022) or noisy two-moons data Scikit-Learn (2022)" (page 1/2)
"genes 10x Genomics (2022) or noisy two-moons data Scikit-Learn (2022)" (page 3)
" e.g. noisy two-moons dataset." (page 3)
Should the authors want to keep these examples, their relevance should be first clearly explained.

4. In Section 2 "Related Work", the cardinality C of the set A need to be clearly defined.
For example, later in the paper C is used to refer to measurements or number of channels, but it would be better to provide a clear yet general definition first and then give specific examples in the current setting (e.g., qMRI application)

5. The subsampling strategy for selecting C out of \bar{C} should be better explained.
For example, JOFSTO is said to "progressively" construct m_t to have C_t ones, but how is this precisely achieved?
Does each step t -> t+1 reduce the C_t by a fixed number? (e.g. removal of a single feature) Or can it be variable?

6. Furthermore, it is not clear how the feature scores quantify the relative importance of the available features and rank them.
This is central to JOFSTO to improve ED performance, and this needs to be clearly elaborated.
Especially, clearly explanation is needed regarding how the task at hand informs this scoring process to make the joint feature selection and task optimization work.

7. On page 2, it is ambiguous what the authors are referring to as "circularity" and how the proposed subsampling-task paradigm for ED avoid this issue.
Please clarify.

8. While the authors mention "studying the size and diversity of densely sampled data needed to ensure strong generalizable design" as future work, there should be at least some empirical evaluations on the minimum C needed to attain a certain level of performance for the task at hand.
Let's denote this C as C_min.
It would be meaningful to understand the performance of JOFSTO (and any under/overfitting of the optimized network models) when C (<C_min) is much smaller than needed or when C (>C_min) is unnecessarily large, at least empirically.
Based on this, prescribing a general guideline for deciding the optimal value of C would be practically important.





**Summary Of The Paper:**

This paper proposes a new method for "experiment design (ED)" by jointly performing feature selection and task optimization.
The performance of the proposed method, called JOFSTO, has been evaluated based on applications in Quantitative Magnetic Resonance Imaging (qMRI), where the evaluation results show that JOFSTO outperforms other existing schemes/baselines under a number of scenarios.



**Summary Of The Review:**

In this work, they propose a new method called JOFSTO for experiment design (ED) (mainly in qMRI), which results in improved parameter estimates and reconstruction results through joint feature selection and task optimization.
The proposed method is simple yet reasonable, and although the work is somewhat limited in terms of the novel methodological contributions it is making, performance assessment results for several qMRI applications show that JOFSTO outperforms other existing schemes/baselines, and hence may provide practical advantages over alternatives.

---

> ### Author Response · Authors · 2022-11-18
> **Response to Reviewer S5n8 part 1 - Detailed Feedback**
>
> We thank RS5n8 for taking the time to provide extensive feedback.
>
> > unclear whether JOFSTO would be applicable to other experiment design problems beyond qMRI applications considered in this study. \
> 1 - proposed work is focused on a very specific application and problem setting, this should be more clearly reflected in the title and abstract.
>
> We appreciate the reviewer's view, since we only included results from one application area, but the proposed method certainly has wider applicability, as it makes no assumption about the data and may be applied to other problems in supervised feature selection.  We have now included two additional results on a distinct application: hyperspectral imaging in section 4.2. We also discuss additional application areas, specifically neuropsychological questionnaire design, in section 5.
>
> > 2 - '"ED in qMRI" seems to be quite different from what is typically referred to "experimental design" or "optimal experiment design (OED)" in various other fields [provided references], ... important to provide a more general treatment of the OED problem in the introduction
>
> ED in qMRI is not substantially different to other scenarios, although it has its particular constraints. We certainly agree that the description should start with a general treatment of OED the specialise to particular applications. This is precisely what we aimed for in the submission. The opening paragraph of Related Work Section 2 (entitled "Experiment Design") is general. The general paragraph already included several of the references the reviewer provides; we thank them for the additional pointers. We note however that all those refs refer to a particular ED paradigm (Bayesian); our general paragraph considers a wider set and our paper introduces a new paradigm.
>
> > 3 - mention certain problems/applications - seemingly not closely related to the current work - without providing any context ...  "e.g. 10x Genomics (2022) ... Scikit-Learn (2022) ... relevance should be first clearly explained
>
> The intention was to illustrate the difference between problems that feature selection algorithms are commonly designed to address with the experiment-design problem we focus on here. We altered pg 1 to emphasize that state-of-the-art techniques in supervised FS are designed to identify a small, highly discriminative subset, and are deployed on those (and similar) datasets.
>
> > 4 - 'Section 2 "Related Work", the cardinality C of the set A need to be clearly defined'
>
> Done, we altered the text.
>
> > 5 - subsampling strategy for selecting C out of $ \bar{C} $ should be better explained. ... JOFSTO is said to "progressively" construct $ m_t $ to have $ C_t $ ones, but how is this precisely achieved? Does each step $ t -> t+1 $ reduce the $ C_t $ by a fixed number? (e.g. removal of a single feature) Or can it be variable?
>
> Fixed, we altered the methods to explain the integers $ C_{t}, t=1,...,T $ are user-chosen hyperparameters.  At the beginning of step $ t $ $ m=m_{t-1} $ has $ C_{t-1} $ ones and $ \bar{C} - C_{t-1} $ zeros.  In  At the end of step $ t $, $ m = m_{t} $ has $ C_{t} $ ones and $ \bar{C} - C_{t} $ zeros.  Details on setting elements of m from one to zero are in supplementaries section A.
>
> > 6  - not clear how the feature scores quantify the relative importance of the available features and rank them ...  how the task at hand informs this scoring process to make the joint feature selection and task optimization work
>
> We altered clarified the results section.  At the end of step $ t $, $ \bar{s}\_{t} \in \mathbb{R}_{+}^{\bar{C}} $ scores each of the $ \bar{C} $ features, higher-scored features have greater importance and are higher ranked.  The feature scores $ \bar{s}_t $ are averages of learnt scores during our end-to-end training, where they are adjusted gradually to promote features/measurements that lead to higher performance in the task network.
>
> > 7 - On page 2, it is ambiguous what the authors are referring to as "circularity" and how the proposed subsampling-task paradigm for ED avoid this issue
>
> We added an additional example, second paragraph of section 1.
>
> > 8 - 'some empirical evaluations on the minimum C needed to attain a certain level of performance for the task at hand ... prescribing a general guideline for deciding the optimal value of C would be practically important'
>
> Thank you for this thought.  We certainly intend to perform this analysis in the future, but it needs careful attention using both simulations and sufficiently rich real data sets. We have not had time to perform it, but have extended the discussion section 5 on this matter to highlight its importance for future work.

---

> ### Author Response · Authors · 2022-11-18
> **Response to Reviewer S5n8 part 2 - Novelty**
>
> > Methodological novelty is somewhat limited
>
> The key novelty lies in the new approach to a long-standing paradigm. The technical novelty over existing feature selection algorithms may be relatively simple, but is highly effective and essential for the novel application we enable.  Nevertheless, we do feel that important technical innovations have been made.  Our approach to perform concurrent feature scoring, feature subsampling, and task-prediction is novel.  Our end-to-end differentiable dual-network that scores features and then performs a task is novel.  Our specific optimization procedure that scores features to then progressively sample the features, and iteratively to drive the subsampling, is also novel.  The novel features of our method is the reason for the 'Compelling performance against strong and recently published baselines' (Reviewer SWxw) on four datasets.

---

### Author Response · Authors · 2022-11-18
**Response to All Reviewers**

We thank the reviewers for their time and valuable feedback.  The key criticism they raise is around generality of the method, since we only demonstrate in qMRI applications.  The intention was to show the diversity of design-execution tasks our new paradigm can enable even within a single application area. However, we appreciate this made the remit appear narrow. We now include two experiments from a distinct application area (hyperspectral imaging in section 4.2) and discuss in more detail other scenarios where JOFSTO offers benefit in section 5.  We have also refined the methodological description to highlight the novelty and motivation from information in the supplementaries.

---

### Decision · Program_Chairs · 2023-01-20

**Decision:**

Reject

**Justification For Why Not Higher Score:**

The paper requires significant revision to better describe their approach and additional experiments regarding requirements on size and diversity of the densely sampled data.

**Justification For Why Not Lower Score:**

N/A

**Metareview: Summary, Strengths And Weaknesses:**

The paper proposes an approach performing joint supervised feature selection and model estimation for experiment design in settings where the training data is densely sampled over the space of possible measurements and a gold-standard output is provided for the task driving the experiment design. The key target application is that of quantitative imaging and the approach is shown to perform very well on such settings.

Strengths: The reviewer and AC all agree that the proposed approach is promising and achieves very nice results on the target applications, and also appreciate the contribution of tackling ED through the lens of FS.

Weaknesses:
- However, while ICLR certainly welcomes imaging oriented papers such as [R1] mentioned by the authors, the present papers requires significant revision to describe the methodology more formally (see e.g. Section 3 of [R1]). This is critical not only for clarity but also to improve significance beyond the target application.
- As mentioned by reviewer S5n8,  it is important to study how the performance evolves w.r.t to C and the diversity of the data and characterize any phase transition etc, and the AC agrees with the reviewer that this should not be relegated to future work.

[R1] Zero-Shot Self-Supervised Learning for MRI Reconstruction (ICLR 2022)